# Uncertainty in the association between socio-demographic characteristics and mental health

Nataliya Rybnikova[1,2], Dani Broitman[1]*, Murielle Mary-Krause[3], Maria Melchior[3], Yakov Ben-Haim[4]

1 Faculty of Architecture and City Planning, Technion—Israel Institute of Technology, Haifa, Israel,
2 Department of Geography and Environmental Studies, Haifa University, Haifa, Israel, 3 Sorbonne Université, INSERM, Institut Pierre Louis d'Epidémiologie et de Santé Publique (IPLESP), Equipe de Recherche en Epidémiologie Sociale (ERES), Paris, France, 4 Faculty of Mechanical Engineering, Technion —Israel Institute of Technology, Haifa, Israel

* danib@technion.ac.il

## Abstract

Questionnaires are among the most basic and widespread tools to assess the mental health of a population in epidemiological and public health studies. Their most obvious advantage (firsthand self-report) is also the source of their main problems: the raw data requires interpretation, and are a snapshot of the specific sample's status at a given time. Efforts to deal with both issues created a bi-dimensional space defined by two orthogonal axes, in which most of the quantitative mental health research can be located. Methods aimed to assure that mental health diagnoses are solidly grounded on existing raw data are part of the *individual validity axis*. Tools allowing the generalization of the results across the entire population compose the *collective validity axis*. This paper raises a different question. Since one goal of mental health assessments is to obtain results that can be generalized to some extent, an important question is how robust is a questionnaire result when applied to a different population or to the same population at a different time. In this case, there is *deep uncertainty*, without any a priori probabilistic information. The main claim of this paper is that this task requires the development of a new *robustness to deep uncertainty axis*, defining a three-dimensional research space. We demonstrate the analysis of deep uncertainty using the concept of robustness in info-gap decision theory. Based on data from questionnaires collected before and during the Covid-19 pandemic, we first locate a mental health assessment in the space defined by the *individual validity axis* and the *collective validity axis*. Then we develop a model of info-gap robustness to uncertainty in mental health assessment, showing how the *robustness to deep uncertainty axis* interacts with the other two axes, highlighting the contributions and the limitations of this approach. The ability to measure robustness to deep uncertainty in the mental health realm is important particularly in troubled and changing times. In this paper, we provide the basic methodological building blocks of the suggested approach using the outbreak of Covid-19 as a recent example.

**Data Availability Statement:** Research participants were guaranteed that the raw data they provided will remain confidential. To request access to the data, please send an email to cohorte.

tempo@inserm.fr. Anonymized data can only be shared after explicit approval of the French national committee for data protection for approval (Commission Nationale del'Informatique et des Liberte´s, CNIL).

**Funding:** This paper is a research output of the project "COVID-19 and Mental Health – dealing with short and long-term Uncertainty" (COMHU). COMHU is a bi-national research project funded by the Ministry of Science and Technology (MOST) of the State of Israel, the Ministry of Europe and Foreign Affairs (MEAE) and the Ministry of Higher Education, Research and Innovation (MESRI) of France. The authors thank the funding institutions for their support. The funders had no role in study design, data collection and analysis, decision to publish, or preparation of the manuscript.

**Competing interests:** The authors have declared that no competing interests exist.

# 1 Introduction

Health assessment is a contested topic in social and epidemiological sciences. In the past, the measurement of health was a combination of quantitative indicators at macro and micro levels (health care expenditure or mortality rates, as an example of the former, clinical data, of the latter). The concept gradually evolved towards the inclusion of qualitative or subjective indicators, such as self-reported health information [1]. The modern measurement of mental health is intrinsically linked to the progressive acceptance of self-reported indicators using standardized questions, which can be traced back to the broad definition of health in the constitution of the World Health Organization from 1946 [2].

However, this type of indicator entailed a range of new questions related to socio-cultural and personal biases [3]. Efforts to define strong links between symptom questions and the diagnosis of disorders are one of the two main axes of quantitative mental health assessment. We call it the *individual validity axis*. Part of the efforts along this axis advocate homogenization and standardization, focusing on the general population or even international comparisons [4–6]. Other studies, on the contrary, develop mental health screening tools tailored to specific populations [7–10]. Research methods located along the individual validity axis aim to assure that mental health diagnoses are solidly grounded on the questions included in the questionnaires and the collected responses to them. The second focus of quantitative mental health assessment is the *collective validity axis*. Its purpose is to allow the generalization of the results across the entire population, attenuating the effect of possible individual variations in the collected sample [11]. The range of variations in this axis is also wide, spanning from traditional statistical methods [12, 13] to modern data science approaches [14, 15].

The *individual* and *collective* validity axes are orthogonal in the sense that any quantitative mental health assessment is located somewhere along each one of them, but a change in one of the axes does not necessarily affect the other. Generalized estimation equations (GEE) models, a statistical tool used for the analysis of longitudinal data, can be applied to test the efficiency of multiple measures of Patient Health Questionnaires (PHQ) [16]. Alternatively, the same GEE models are used as well to analyze the mental health impacts of lockdowns through Adult Self Report (ASR), a well-validated instrument to assess adult psychopathology [17].

The *individual* and the *collective* validity axes span a wide and rich bi-dimensional space in which much quantitative mental health research thrives. Indeed, this bi-dimensional space grows steadily with the publication of new quantitative research using validated mental health instruments and analyzed with inference tools. The present paper is not designed to contribute to this type of research. Instead, the intention is to raise questions about uncertainty that cannot be answered in the context of the space spanned by the two mentioned axes. This is the motivation for the suggestion of a third axis, able to provide a response based on info-gap theory [18, 19].

Sample data designed using any of the validated mental health instruments are collected at a specific time, under specific conditions. These data can be analyzed proceeding exclusively from the sample itself. Alternatively, these data can be used to infer statistical results about the general population from which the sample was drawn, if certain probabilistic information is assumed. However, deep uncertainty arises in new and unfamiliar situations, such as different populations or times, and is characterized by the absence or serious deficiency of probabilistic information.

Knight distinguished between "Risk as a known chance and true Uncertainty" which is "unmeasurable" in any probabilistic sense [20, pp.20, 21]. Knightian uncertainty arises from a substantial absence of data, knowledge or understanding relevant to a decision at hand. Knightian uncertainty deals with situations for which probabilities are unknown, so likelihoods cannot be calculated and conventional probabilistic insurance against failure is impossible.

Nonetheless, facing Knightian uncertainty, one can still ask: to what extent can the data change (e.g. by considering a different time or location) and the original conclusions are still valid? The question is not probabilistic. We are not asking: What is the likelihood of validity? The question is: How robust are the conclusions to variation in the data themselves? The concept of robustness developed here is non-probabilistic and is based on info-gap decision theory.

This question of robustness to Knightian uncertainty cannot be answered in the space defined by individual/collective validity. This question requires an additional perspective: robustness to non-probabilistic uncertainty. This perspective can be viewed as an orthogonal axis to the two previous axes in the following sense. Analysis of the robustness of the results does not explicitly influence either the individual or collective validity of the methodology used in the analysis. Instead, analysis of robustness to non-probabilistic uncertainty can be applied to any methodological context composed of the individual/collective validity components. Fig 1 schematically shows a 3D space of the three components.

In epidemiological research, examining the robustness of results can prevent both under- and over-estimating public health risks and help produce more informed public health policies. To the best of our knowledge, research in this field is focused solely on individual validity or collective validity. The present study aims to fill this gap. The present paper aims at developing the methodology of robustness analysis for a pre-chosen research task–examining the associations between socio-demographic characteristics and mental health disorders in an adult French population. We examine these associations for two time periods: before and during the COVID-19 pandemic. By doing so, we aim to examine the robustness of the derived conclusions about the presence or absence of the examined associations both in normal and extreme conditions. We believe that this would contribute to developing more informed differential policies in the field of mental health protection, which would suit specific situations the best.

The paper is structured as follows: Section 2 –the core section of the paper–describes each of the methodological components, with emphasis on the robustness analysis. Section 3 explains the data set and the substantive questions used for the demonstration of the methodology. Section 4 reports the results of the application of the methodology to the substantive questions. Finally, section 5 discusses and summarizes the results, and proposes directions for further research.

## 2 Methodology: Individual, collective, and robustness components

We proceed in three stages. (i) We use the Adult Self Report (ASR) scale evaluating mental health [21, 22] to identify the presence of mental health difficulties before and during the COVID-19 pandemic among study participants (individual validity component). (ii) We use $\chi^2$-analysis to test for statistically significant differences in mental health among different groups of participants (collective validity component) based on estimated probabilities of membership in the groups. (iii) Finally, we employ info-gap theory to assess the robustness of the conclusions to Knightian uncertainty in the data.

### 2.1. Individual validity: ASR-based estimates of mental health and socio-demographic characteristics

In this study, we assume that the individual validity component is given. We use the ASR, which is a validated measure of mental health based on symptoms of depression, anxiety, and somatic disorder. Therefore, we begin by studying the association between socio-demographic characteristics and mental health difficulties. The estimates of mental health are given as dichotomized variables (implying either the absence or presence of certain symptoms) according to the 85th percentile of the ASR scale, representing substantial presence of adverse mental

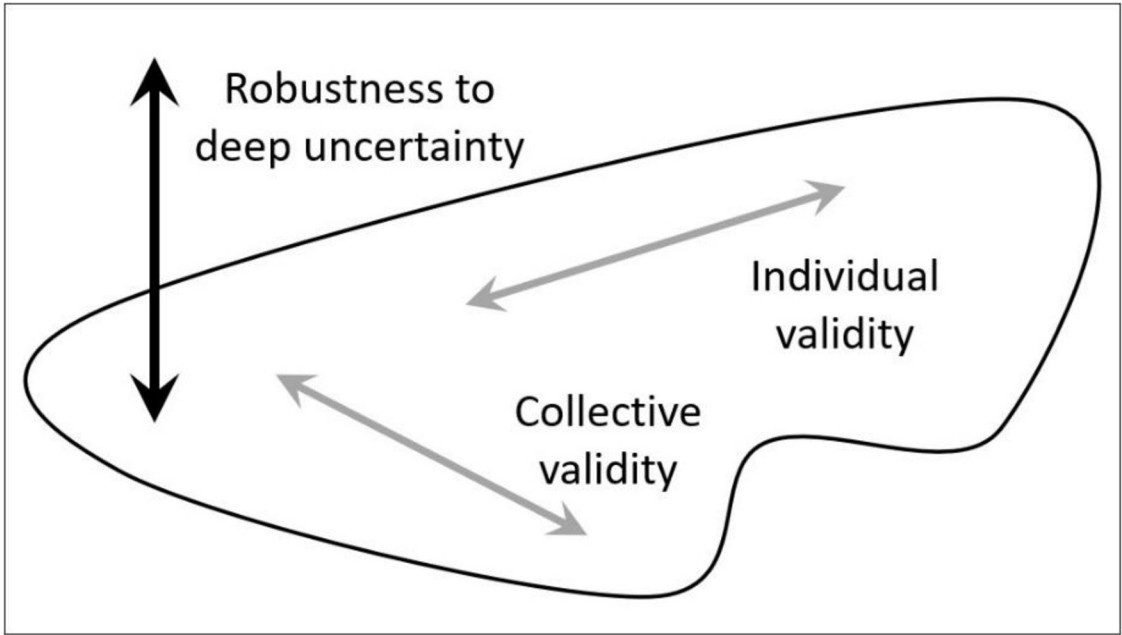

**Fig 1. Geometric metaphor.** A 3D space of research methodology–orthogonal composition of individual validity, collective validity, and robustness of the results.

health symptoms. We count the individuals with and without a particular mental disorder among different socio-demographic groups, also defined as dichotomous. In section 3 we describe the data in detail, but for the purpose of the individual validity description it is enough to stress that in all cases we use two dichotomous variables. Therefore, we obtain a set of 2x2 matrices for further analyses (see Table 1).

## 2.2. Collective validity: $\chi^2$ test for assessing the 'mental health–socio-demographic characteristics' associations

To identify the absence or presence of an association between a socio-demographic characteristic and mental health, we utilize the $\chi^2$ test to compare the marginal probability distributions of the two populations. For each 2x2 matrix (see Table 1 in subsection 2.1), we consider two different populations (the 2 rows); in each population, there are two different types of members (the two columns). Let $p_{ij}$ denote the probability that an observation chosen randomly from the $i^{\text{th}}$ population is of type $j$. This probability distribution is normalized over the two types, so $p_{i1} + p_{i2} = 1$ for $i = 1, 2$. The null hypothesis is that, for each type, $j$, the probabilities $p_{ij}$ are the same in each population, and that the groups and types are all statistically

**Table 1. Mental health model: Numbers of respondents.**

|  | Absence of mental health disorder (j = 1) | Presence of mental health disorder (j = 2) |
| --- | --- | --- |
| Socio-demographic group #1 (i = 1) | $N_{11}$ | $N_{21}$ |
| Socio-demographic group #2 (i = 2) | $N_{12}$ | $N_{22}$ |

independent. The alternative hypothesis is that this is false. Specifically:

$$H_0 : p_{11} = p_{21} \ and \ p_{12} = p_{22}$$
$$H_1 : H_0 \ \text{is false}$$

(1)

Thus $H_0$ is the assertion that the probability of population type $j = 1$ is the same for both populations, $i = 1$ and $i = 2$. Because $p_{ij}$ is normalized over $j$, hypothesis $H_0$ likewise asserts that the probability of population type $j = 2$ is the same for both populations, $i = 1$ and $i = 2$. In other words, we are assessing the association between mental health and socio-demographic characteristics in terms of the probability distributions of these properties.

The emphasis of this paper is primarily methodological, focusing on extending the analysis to consideration of Knightian uncertainty. As a result, we consider only three pairs of population-type and socio-demographic variables, even though there are many possible pairs and the analysis could be extended to larger groups of these variables. Again, the emphasis is on the methodology for extending the analysis to Knightian uncertainty, rather than on a detailed substantive analysis of relations between populations and demographics.

Let $N_{ij}$ denote the number of observations in the sample of type $j$ from population $i$. Let $N_{i+}$ denote the total number of observations in the samples from population $i$ for $i = 1,2$. Likewise, let $N_{+j}$ denote the total number of observations of type $j$ for $j = 1,2$. Finally, let $N$ denote the total number of observations. Specifically:

$$\left. \begin{array}{c} N_{i+} = N_{i1} + N_{i2} \\ N_{+j} = N_{1j} + N_{2j} \end{array} \right\} \Rightarrow N = N_{1+} + N_{2+} = N_{+1} + N_{+2}$$

(2)

Let $\tilde{N}_{ij}$ denote the observed number of observations in the sample of type $j$ from population $i$ while $N_{ij}$ denotes the unknown true value at some future time or place. Let $\tilde{N}_{i+}$, $\tilde{N}_{+j}$ and $\tilde{N}$ denote the observed values of $N_{i+}$, $N_{+j}$ and $N$, respectively.

The $\chi^2$ statistic is

$$Q = \sum_{i=1}^{2} \sum_{j=1}^{2} \frac{\left(N_{ij} - \hat{N}_{ij}\right)^2}{\hat{N}_{ij}}$$

(3)

where we have defined the following empirical approximation, conditioned on $H_0$, for the expected number of observations of type $j$ in group $i$:

$$\hat{N}_{ij} = \frac{\tilde{N}_{i+}}{\tilde{N}} \cdot \frac{\tilde{N}_{+j}}{\tilde{N}} \cdot \tilde{N} = \frac{\tilde{N}_{i+} \cdot \tilde{N}_{+j}}{\tilde{N}}$$

(4)

Conditioned on $H_0$, $Q$ will approximately have a $\chi^2$ distribution (strictly true in the limit of infinite data) with $(2–1)\cdot(2–1) = 1$ degree of freedom. $Q_c(p)$ represents the quantile of the $\chi^2$ distribution corresponding to a cumulative probability of $p$. Specifically, with one degree of freedom, $Q_c(0.95) = 3.84$. If $H_0$ holds, then the probability that the $\chi^2$ statistic exceeds 3.84, just by chance, equals 0.05. Therefore, one rejects the hypothesis $H_0$ at level of significance $p = 0.95$ if $Q > 3.84$. That is, $H_0$ is rejected at level of significance $p$ if:

$$Q > Q_c(p)$$

(5)

Otherwise, $H_0$ is not rejected, and one concludes that there is no significant variation in mental health outcomes among different socio-demographic groups of the population.

## 2.3. Robustness to uncertainty: Info-gapping the $\chi^2$ test

Our analysis is based on the concept of robustness to uncertainty as developed in info-gap theory [18, 19, 23]. The assessment of robustness is based on three components: a system model, a performance requirement, and an info-gap model of uncertainty. Robustness is the greatest horizon of uncertainty (in the uncertainty model) up to which the system model satisfies the performance requirement. In the present application, the system model is the $\chi^2$ statistic, and the performance requirement is that the statistic must not exceed a critical value, for any realization of data in the uncertainty model. This analysis has three distinctive characteristics.

First, uncertainty is represented non-probabilistically. The data upon which the analysis is based are highly uncertain for several reasons in addition to standard statistical variability. The data reflect a specific population, while we may be interested in a different population at some other time or location, or perhaps the original population at a different time. In other words, the data entail what Knight [20] called "true" uncertainty for which probability distributions are lacking. An info-gap model of uncertainty is an unbounded family of nested sets of an uncertain entity. An info-gap model is non-probabilistic and represents Knightian uncertainty.

The second distinctive characteristic of the info-gap analysis of robustness is that the performance is satisfied—satisfying a critical requirement—rather than optimized.

Third, the robustness to uncertainty is optimized. That is, we optimize robustness to uncertainty, to achieve an acceptable but possibly sub-optimal outcome. We will see subsequently that robustness trades off against performance: better performance has worse robustness to Knightian uncertainty. This motivates satisfying the performance and optimizing the robustness.

Recall that $\tilde{N}_{ij}$ denotes the observed number of observations in the sample of type $j$ from population $i$ while $N_{ij}$ denotes the unknown true value in some future time or place. Recall also that $\tilde{N}_{i+}$, $\tilde{N}_{+j}$ and $\tilde{N}$ denote the observed values of $N_{i+}$, $N_{+j}$ and $N$, respectively. We have insufficient information to choose a probability distribution for these uncertain variables in possible future situations. Nonetheless, a rough first approximation is that the observations are statistically independent and identically distributed. This approximation, were it true, would imply that the observations are Poisson random variables. If we treat $\tilde{N}_{ij}$ as an estimate of the mean for that category, then the standard deviation of the observations would be approximately $\sqrt{\tilde{N}_{ij}}$ under the Poisson approximation. We adopt this as an approximation of the relative range of variability of the observations, without assuming that the uncertain future observations are actually Poisson distributed. More precisely, we adopt a fractional-error info-gap model for uncertainty in the observations, in which the range of variation of the unknown true values are centered on the observations, and normalized by $\sqrt{\tilde{N}_{ij}}$. Specifically, the info-gap model for uncertainty in the observations becomes:

$$U(h) = \left\{ N_{ij} : N_{ij} \geq 0, \; \left| \frac{N_{ij} - \tilde{N}_{ij}}{\sqrt{\tilde{N}_{ij}}} \right| \leq h, \; i = 1, 2, \; j = 1, 2 \right\}, \; h \geq 0 \tag{6}$$

The $N_{ij}$ only take discrete values, but we are treating them as continuous variables for simplicity. The maximum distance between a continuous variable and the nearest discrete variable is 0.5. Thus, the continuous approximation of the discrete $N_{ij}$ values entails an error no larger than plus or minus 0.5. The values of the observed variables are typically in the range from 70 to 150,

with the smallest value of 16 and the largest value of 506. The fractional-error normalization in the info-gap model of Eq (6) is $\sqrt{\tilde{N}_{ij}}$ whose smallest value is 4 and with typical values from 8 to 12. The errors resulting from the continuous approximation are thus small, and this approximation has negligible impact on the evaluation of the robustness to uncertainty of this data.

The set $U(h)$ contains all data sets, $N_{ij}$, where values deviate fractionally from the corresponding observed values, $\tilde{N}_{ij}$, by no more than the value $h$. We make no assumptions about the relative likelihood of possible realizations of $N_{ij}$. The info-gap model in Eq (6) presumes no specific probability distribution. We used the Poisson concept to choose the normalization of the uncertainty-range for each $N_{ij}$. Furthermore, the value of $h$ is unknown, and the sets $U(h)$ become more inclusive as $h$ grows, and there is no probabilistic normalization of the likelihood of the elements of the sets $U(h)$. This unboundedness endows $h$ with its meaning as an horizon of uncertainty. The info-gap model is thus not a single set, but rather a family of nested sets that is unbounded in the space of possible data sets. The info-gap model represents uncertainty (in the data) without any probabilistic model for likelihood of future observations. In other words, the info-gap model of uncertainty is a quantification of Knightian true uncertainty.

Let us suppose that the observations are such that $H_0$ is not rejected at significance level $p$ based on the observations. That is, the observed value of the $\chi^2$ statistic, $Q$ in Eqs (3) and (4) based on the observed values $\tilde{N}_{i+}$, $\tilde{N}_{+j}$ and $\tilde{N}$, does not exceed the value $Q_c(p)$. The robustness for not rejecting $H_0$ is the greatest horizon of uncertainty $h$, up to which Eq (5) does not hold for all values of the $\chi^2$ statistic, $Q$, based on all observations in the info-gap model of Eq (6):

$$\hat{h}_0(p) = \max\left\{ h : \left( \max_{N_{ij} \in U(h) \forall i,j} Q \right) \leq Q_c(p) \right\} \tag{7}$$

$\hat{h}_0(p)$ is defined as zero if this set of $h$ values is empty. Reading this relation from left to right: the robustness for accepting $H_0$ at level of significance $p$ is the greatest horizon of uncertainty, $h$, up to which the $\chi^2$ statistic, $Q$, is no greater than the critical value, $Q_c(p)$, for all possible realizations of the data in the uncertainty set $U(h)$.

An analogous definition of robustness holds when $H_0$ is rejected at level of significance $p$ based on the observations. Specifically, the robustness for rejecting $H_0$ at level of significance $p$ is the greatest horizon of uncertainty, $h$, up to which Eq (5) holds for at least one value of the $\chi^2$ statistic, $Q$, based on observations in the info-gap model of Eq (6):

$$\hat{h}_1(p) = \max\left\{ h : \left( \max_{N_{ij} \in U(h) \forall i,j} Q \right) > Q_c(p) \right\} \tag{8}$$

$\hat{h}_1(p)$ is defined as zero if this set of $h$ values is empty. It is evident that $\hat{h}_0(p)$ and $\hat{h}_1(p)$ cannot both be greater than zero, though both are non-negative.

Consider the constraint on $Q$ in the inequality in Eq (7), this constraint is relaxed as the value of the critical statistic, $Q_c(p)$, increases. The robustness is the greatest horizon of uncertainty up to which this constraint is not violated. As a result, the robustness function in Eq (7), thought of as a function of the critical statistic $Q_c(p)$, increases as $Q_c(p)$ increases. In other words, the robustness is a monotonic function: The robustness increases as $Q_c(p)$ increases. The robustness function is thus fully characterized by its inverse function because the robustness is monotonic. Consequently, the robustness function in Eq (7) can be evaluated by evaluating its inverse function.

Let $m_0(h)$ denote the inner maximum in Eq (7). A plot of $h$ vs. $m_0(h)$ is identical to a plot of $\hat{h}_0(p)$ vs. $Q_c(p)$. That is, $m_0(h)$ is the inverse function of $\hat{h}_0(p)$. This inverse function is evaluated as a constrained optimization. Specifically, maximize $Q$, as defined in Eqs (3) and (4),

subject to the following constraints:

$$\forall i, j \begin{cases} N_{ij} \geq 0 \\ N_{ij} \leq \tilde{N}_{ij} + h \cdot \sqrt{\tilde{N}_{ij}} \\ N_{ij} \geq \tilde{N}_{ij} - h \cdot \sqrt{\tilde{N}_{ij}} \end{cases} \tag{9}$$

Similarly, the robustness function in Eq (8) is evaluated by evaluating its inverse function. Let $m_1(h)$ denote the inner minimum in Eq (8). A plot of $h$ vs. $m_1(h)$ is identical to a plot of $\hat{h}_1(p)$ vs. $Q_c(p)$. That is, $m_1(h)$ is the inverse function of $\hat{h}_1(p)$. This inverse function is evaluated also as a constrained optimization, that is, minimize $Q$, as defined in Eqs (3) and (4), subject to the constraints defined in Eq (9).

Finally, consider the specific cases of Eqs (7) and (8) in which $Q_c(p) = 3.84$ which is the $\chi^2$ threshold for $p = 0.95$. In these cases, $\hat{h}_0(p)$ and $\hat{h}_1(p)$ are called "critical robustnesses". This means that if either critical robustness is large, then the corresponding hypothesis can be confidently adopted for a wide range of Knightian uncertainty. Conversely, a small value of a critical robustness implies that the corresponding hypothesis is vulnerable to even a low level of Knightian uncertainty in the data. The values of the critical robustness are specified in the caption of Table 3, and represented by vertical dotted lines in the graphs presented below.

## 3 Substantive mental health questions

### 3.1. Motivation

Past research on the long-term psychological consequences of an epidemic outbreak was based on infectious disease spreads which were concentrated in specific geographic areas, and where quarantine measures, when applied, were limited to specific groups [24, 25]. Moreover, with important exceptions, most past studies were cross-sectional or qualitative, and the role of pre-existing psychological difficulties could not be fully explored [26]. It is difficult to know the extent to which the results of these previous studies apply to the Covid-19 virus outbreak, where entire populations of many countries were required to implement social distancing measures for preventive purposes. Short-term effects of the pandemic are only starting to be unveiled, both regarding mental health assessments [27], and the impacts on mental health care [28, 29]. The impact of the Covid-19 pandemic and of universal quarantine and social distancing measures that lasted several weeks, on long-term patterns of mental health, are yet unknown.

A variety of studies examining the influence of the pandemic on mental health across different countries and population groups has emerged since 2020. Table 2 summarizes the findings on the associations between individual characteristics and mental health problems revealed by these studies.

As can be seen in Table 2, some individual characteristics appear to be associated with higher rates of mental health problems. However, the results vary greatly. As shown by a recent review, the prevalence of mental health problems in the context of the COVID-19 pandemic varied from 6.33% to 50.9% for symptoms of anxiety, from 14.6% to 48.3% for symptoms of depression, and from 8.1% to 81.9% for symptoms of stress [39]. This means that the findings of a selected study, conducted on a certain population, might not be applicable to other populations (even within the same country), let alone other countries and regions.

Therefore, we apply the proposed methodology to the analysis of associations between mental health disorders and individual characteristics in the adult population in France. First, we are interested in assessing the degree to which associations between socio-demographic

**Table 2. A summary of several recent studies on individual characteristics and mental health during the course of the COVID-19 pandemic (selected references retrieved from a search in google scholar using the keywords "Covid-19" and "mental health" in December 2022).**

| Population examined | Number of participants | Pre-pandemic analysis included | Mental health problems analyzed | Young people | Female gender | Living with children | Lower income | Living alone | Poor health status | Student status | Lower education | Reference |
|---|---|---|---|---|---|---|---|---|---|---|---|---|
| Adult population of the UK | 17,452 | Yes | Mental health | + | + | + | | | | | | [30] |
| French students in Eastern France | 8,004 | No | Depression | | + | | + | | | | | [31] |
| | | | Anxiety | | + | | + | | | | | |
| | | | Distress | | + | | + | + | | | | |
| Adult population of China | 1,210 | No | Stress, anxiety, depression | | + | | | | + | + | | [32] |
| Adult population of England | 36,520 | No | Anxiety and depression | + | + | + | + | + | | | + | [33] |
| Adult population of France | 1,224 | Yes | Anxiety and depression | | + | | + | + | | | | [17] |
| Adult population of Denmark | 8,261 | No | Loneliness, anxiety, mental health scale | + | + | | | | | | | [34] |
| Adult population of Denmark, France, and UK | 69,136 | No | Loneliness, life satisfaction | | | | | + | | | | [35] |
| Adult population of Lybia | 8,084 | No | Anxiety | | + | + | + | + | + | | | [36] |
| Adult and young population in China | 1,074 | No | Anxiety, depression and substance use | + | + | + | | | | + | | [37] |
| Adult population of Italy | 1,177 | No | Depression, Unworthiness, Alienation, Helplessness | | + | + | + | + | | | | [38] |

characteristics and mental health disorders have changed as a result of the COVID-19 pandemic. Then we calculate the robustness-to-uncertainty of these conclusions using an info-gap analysis of all the associations under study.

## 3.2. Data set

Data used in the present analysis are derived from the TEMPO (Trajectoires EpideMiologiques en POpulation) and TEMPO COVID-19 project. The latter is an extension of the existing cohort study, TEMPO, which aims to evaluate individual, familial, and social determinants of mental health difficulties, addictive behaviors, and their trajectories over time [40]. The TEMPO COVID-19 project comprises 9 waves of data collection, online or via postal questionnaires, performed between March 2020 and May 2021.

For the purpose of this analysis, we used mental health outcome variables assessed using the Adult Self Report (ASR) Achenbach System [21, 22]. This scale is an adult extension of the widely used Child Behavioral Checklist (CBCL). Subscales in the present analysis include 18 items to evaluate symptoms of anxiety and depression. Each item was scored 0, 1 or 2, and the level of difficulty was calculated by summing up all relevant items. Afterward, for inter-questionnaire comparison, ASR scores were standardized from 0 to 100. Finally, the scores were dichotomized using the 85th percentile, and only participants with higher scores were considered as having mental health difficulties (anxiety, depression, and somatic disorders).

To assess participants' socio-demographic characteristics we used respondents' gender (male or female), marital status (single or non-single), and living together with at least one child (yes or no) available from the study questionnaires.

**Table 3. Associations between selected participants' socio-demographic characteristics and mental health difficulties that change between pre- and pandemic periods.** The first row reports $\chi^2$ statistics (values exceeding the critical value of 3.84 are marked in bold font), and the 2nd row the critical robustness for accepting/rejecting $H_0$.

| Socio-demographic characteristic | | | Anxiety symptoms | | Depression symptoms | | Somatic symptoms | |
|---|---|---|---|---|---|---|---|---|
| | | | Q0 | Q1* | Q0 | Q1* | Q0 | Q1* |
| 1 | Gender | $\chi^2$ statistic | | | **18.52** | **22.77** | | |
| | | Critical robustness | | | 1.40 | 1.75 | | |
| 2 | Marital status | $\chi^2$ statistic | 3.23 | **5.92** | | | | |
| | | Critical robustness | 0.05 | 0.20 | | | | |
| 3 | Living with a child | $\chi^2$ statistic | | | | | **9.82** | 0.74 |
| | | Critical robustness | | | | | 0.55 | 0.55 |

In the study, we used a prior TEMPO questionnaire performed in 2018 for the pre-pandemic analysis. To analyze the associations during the pandemic period we have aggregated nine waves of data collection within TEMPO COVID-19, performed between March 2020 and May 2021. The sizes of the aggregated pandemic-time sample emerged comparable with those of the pre-pandemic samples. Specifically, the sample sizes for pre-pandemic-time varied from 829 to 858 respondents, depending on the combinations of mental health difficulty and socio-demographic characteristic. For the pandemic-time questionnaires, the corresponding numbers were from 654 to 731.

# 4 Results

## 4.1. Chi-square analyses

Table 3 reports selected $\chi^2$ statistic values for testing associations between participants' socio-demographic characteristics prior to the COVID-19 pandemic (questionnaire Q0) and during the COVID-19 pandemic (aggregated questionnaire Q1*). The selection was based on two types of cases: switching conclusions between pre-pandemic and pandemic times (absence of association before but presence afterwards, or the opposite) or cases in which $H_1$ is accepted in both time periods. We selected only one socio-demographic characteristic that exemplifies each type of case. Other results, namely, lack of association during both periods, are not reported.

The relation between gender and depression is an example of a stable association *regardless of the Covid-19 pandemic*. This means that both before and after the pandemic there is a significantly different outcome regarding depression in men and women. This is demonstrated by consistent $\chi^2$ statistic values well above the critical value of 3.84 as seen in row 1.

But there are also associations that changed during the pandemic. An example of this switch is the 'marital status–anxiety disorder' reported in row 2: The $\chi^2$ statistic switches from a value below the 3.84 threshold before the pandemic, to a value above the threshold during the pandemic, so the decision *switches from $H_0$ to $H_1$*. This means that, even if before the pandemic there were no significant differences between married or single persons regarding the level of reported anxiety, significant differences emerged after the Covid-19 outbreak.

A *third* type of pre- and post-pandemic mental health change is exemplified by the association 'living with a child–somatic disorder' (row 3 in Table 3). An opposite trend is observed here: Before the pandemic, somatic disorders had significantly different impact among persons living with children or in households composed of adults only ($\chi^2$ well above 3.84). This supports hypothesis $H_1$. However, during the pandemic period, these different impacts seem to

vanish, as evidenced by a drop of the $\chi^2$ statistic to 0.74. This result leads to the conclusion that $H_0$ (no significant difference) should be accepted during the pandemic.

## 4.2. Info-gap analysis of robustness

These conclusions are drawn from data obtained from specific populations at specific times. We use the concept of robustness to Knightian uncertainty, as developed in info-gap theory, to assess the degree of confidence one has in applying these conclusions to other populations or other times [18, 19]. The analysis is summarized in Fig 2 (below), that corresponds to each of the cases described in Table 3. We now discuss those graphs.

Fig 2(1) depicts the association between gender and depression. The $\chi^2$ statistic reported in Table 3 indicates that the chance of experiencing depression is not the same for persons of different gender. $H_0$ is rejected with $\chi^2$ value of 18.52 before Covid-19, and 22.57 afterwards. These values are represented in the graph by the intersection of the black and red curves, respectively, with the horizontal axis (the intersection at 22.57 lies outside the graph). The robustness on the horizontal axis is precisely zero. This means that the observed values, 18.52 and 22.57, have no robustness against change in the population. In other words, with zero robustness, we have no confidence in the generality of these specific numerical results. This is the zeroing property of robustness.

But let's look at the overall behaviour of both curves in Fig 2(1). The question that the info-gap analysis addresses is the following: To what extent would we be confident in the conclusions (i.e., the presence of the examined association both before and after the pandemics) if these questionnaires were used in different locations or at different times? In other words, to what extent can the data underlying row 1 in Table 3 vary while also yielding the same conclusions? If the same conclusions are inferred even from very different data sets, then the inference is plausible for very different circumstances. Conversely, if even small data variations alter the conclusions, then the inference is implausible under different circumstances.

The black and red lines in Fig 2(1) represent the info-gap robustness of the observed results to Knightian uncertainty for the two $\chi^2$ tests discussed above. The horizontal axes in the figure depict the critical value of $\chi^2$, namely $Q_c(p)$, while the vertical axes show the robustness to uncertainty, $\hat{h}_0(p)$ or $\hat{h}_1(p)$. The negative slopes of these robustness curves express the irrevocable trade off between robustness to uncertainty and critical value: greater critical value (more definitive statistical decision) entails lower robustness against Knightian uncertainty.

The vertical grey line ($Q_c(p) = 3.84$) is the statistical decision threshold for $p = 0.95$: Cases to the left indicate accepting $H_0$ at 0.95 significance, while cases to the right indicate rejecting $H_0$ at 0.95 significance. Both robustness curves in Fig 2(1) increase from right to left. The black curve (before the pandemic) intersects the threshold at robustness of 1.40 and the red curve (during the pandemic) at 1.75. These values of robustness mean that, for variation of the data up to 1.40 standard deviations in the pre-pandemic data (black curve), or up to 1.75 standard deviations for the pandemic data (red curve), the upper boundary of the range of the calculated $\chi^2$ statistic would not exceed the critical value of 3.84 (vertical grey line): One can maintain that the rejection of $H_0$ holds within these intervals. But if the divergence of the data goes further and exceeds 1.40 or 1.75 standard deviations (for the black and red curves respectively), the maximal value of the calculated $\chi^2$ can exceed the threshold ($Q_c(p) = 3.84$) and the rejection of $H_0$ is no longer justified for populations or times other than those observed. In both cases, the confidence in generalizing these results to other populations or times is moderate.

Fig 2(2) displays the robustness curves for the relation between marital status and anxiety. These curves show a switch in the relation between pre-pandemic and pandemic times. The association between marital status and anxiety was not significant before the pandemic ($\chi^2$

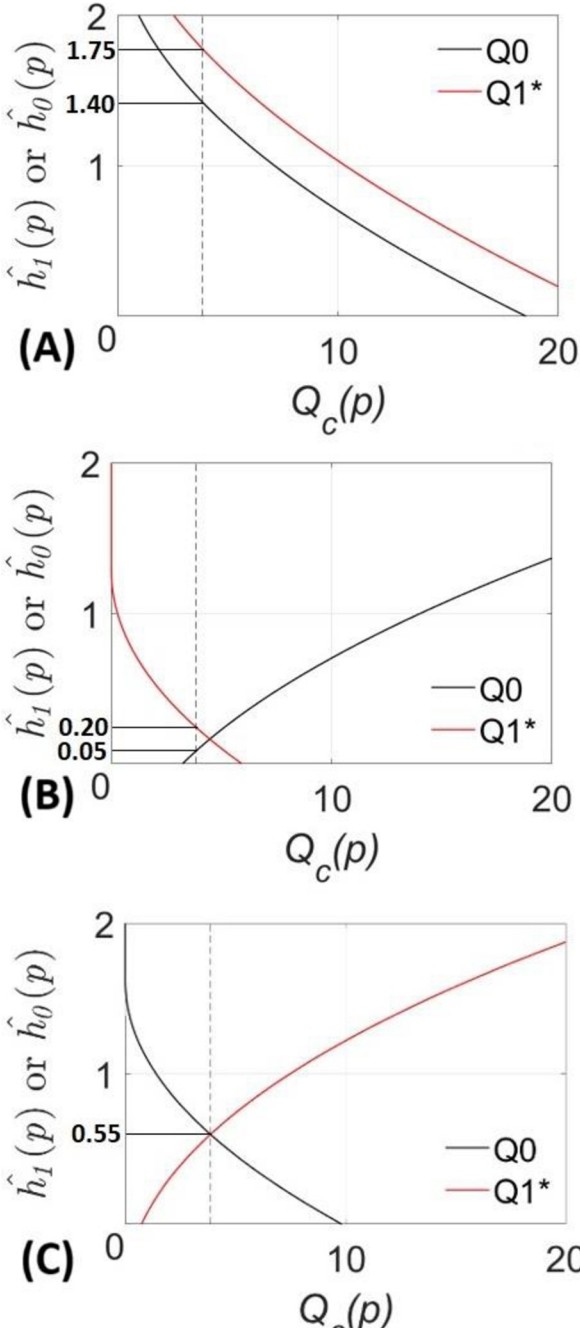

**Fig 2. Robustness functions.** Robustness functions of the associations between the Socio-demographic characteristics and mental health conditions described in Table 3. (A) Gender–Depression. (B) Marital status–Anxiety. (C) Living with a child–Somatic.

statistic was 3.23; horizontal axis of the black curve) but becomes significant during the pandemic ($\chi^2$ statistic of 5.92; red horizontal axis). However, these horizontal intercepts are subject to the zeroing property of robustness: We have no confidence in obtaining precisely the same results in different populations or times. As we allow the values to diverge from the observed values, the black curve (accepting $H_0$, like in Fig 2(1)) increases from left to right, and the red

curve increases from right to left (rejecting $H_0$). This time the question is to what extent would we be confident in these conclusions (i.e., on the absence of the examined association before the pandemic and the presence of it afterwards) if these questionnaires were used in different locations or at different times. The response is given, again, by the intersection of both curves with the threshold value of 3.84. The conclusion that there is no association between marital status and anxiety before the pandemic (i.e., black curve, acceptance of $H_0$) holds only if the sample variation does not exceed 0.05 standard deviations. On the other hand, the conclusion that there is association between marital status and anxiety during the pandemic (i.e., red curve, rejection of $H_0$) holds only if the sample variation does not exceed 0.20 standard deviations. In both cases, this is a quite low robustness and we have little confidence in the generalizability of these results to other circumstances.

The black and red lines in Fig 2(3) are info-gap robustness curves for the observed results to Knightian uncertainty in the second case of switching $\chi^2$ statistics. Living with a child was significantly associated with somatic symptoms before the pandemic, but the association vanished during the pandemic. Again, the horizontal axis in Fig 2(3) depicts the critical value of $\chi^2$, while the vertical axis shows the robustness to uncertainty. The vertical grey line ($Q_c(p) = 3.84$) is, as previously, the statistical decision threshold: Cases to the left indicate accepting $H_0$, while cases to the right indicate rejecting $H_0$. The black curve (before the pandemic) intersects the horizontal axis at 9.82 (rejecting $H_0$) while the red curve (during the pandemic) intersects it at 0.74 (accepting $H_0$). This is the third example of the zeroing property of robustness: these observed values have no robustness to change in the data. We also see trade off between the robustness and the critical value, as expressed in the slopes of the robustness curves. The black robustness curve increases from right to left indicating that less definitive rejection of $H_0$ has greater robustness to Knightian uncertainty. The red curve increases from left to right indicating that less definition acceptance of $H_0$ has greater robustness. In this particular example both curves intersect the threshold (Q = 3.84) at the same robustness of 0.55. This is interpreted as in the previous cases: If the sample before the pandemic varies up to 0.55 standard deviations (but not more), the conclusion of a significant association still holds. Likewise, if the sample after the pandemic varies up to 0.55 standard deviations, the lack of association between living with a child and somatic symptoms (i.e., accepting $H_0$) still holds. In both cases the robustness to Knightian uncertainty is low.

## 5 Discussion and Conclusions

### 5.1. Results related to the mental health impact of the Covid-19 pandemic

The present study develops a methodology for assessing the robustness-to-uncertainty of results from a selected epidemiological analysis. As a case study, we used a data set of self-reports on mental health and socio-demographic characteristics of an adult population in France before and during the COVID-19 pandemic. $\chi^2$ analysis was chosen for examining the absence or presence of the associations between different mental health disorders and socio-demographic characteristics of the respondents. The focus of the paper is the development of a methodology of assessing *the robustness of the conclusions to non-random variation in the data*, specifically, assessing the confidence in applying the conclusions to other populations or times.

The analysis revealed three types of relevant associations between socio-demographic characteristics and mental health when comparing the period before the pandemic and after the outbreak. The first type refers to stable associations both in pre-pandemic and pandemic periods. This is the case for the association between gender and depression which occurs both before the pandemic and after its outbreak (row 1, Table 3). This finding is in line with previous studies reporting the existence of associations between mental health problems and gender

during the pandemic [17, 30–38]. Compared with these studies, our results indicate that these associations hold for pre-pandemic times too.

The second type of association between socio-demographic characteristics and mental health refers to a lack of association before the pandemic and the emergence of association after the Covid-19 breakout for some characteristics under study. This is the case for associations between marital status and anxiety (row 2, Table 3). These results support previous findings about the existence of an association between living alone and higher rates of anxiety disorder [31, 33, 36–38].

Finally, the third (and perhaps least expected) type of association between socio-demographic characteristics and mental health refers to the disappearance of significant associations during the pandemic, despite their existence in the previous period. This is the case for 'living with a child–somatic disorder' (row 3, Table 3). We can only speculate about the meaning of the fading out of these associations. One possible explanation may be the "democratic effect" of the pandemic: Certain characteristics that in normal times sharply differentiate between population groups (e.g. having or not having children) may be less significant in times of emergency [36–38, 41, 42]. An additional "democratic effect" may be the temporary resurgence of the welfare state and grassroots social involvement during the pandemic [43–45].

## 5.2. Results related to the robustness to deep uncertainty

The paper adds a new dimension to traditional quantitative mental health research. In addition to the *individual* and the *collective* validity axes, we introduce the *robustness to deep uncertainty* axis. Implementing this concept by means of info-gap decision theory, we are able to quantify deep Knightian uncertainty without any a priori probabilistic information. A three-dimensional research space is defined by introducing this new axis. Within this three-dimensional space we explore to what extent the conclusions drawn from the data can be reliably applied to future populations in different situations.

The results indicate that in most of the cases the robustness to uncertainty is quite low. Only in a few cases the robustness surpasses the value of 1, and the maximal observed robustness value is 1.75, which is still only moderate robustness as explained earlier. Low robustness values such as these indicate that relatively small changes in numerical results (deriving from samples of different populations or times) are sufficient to invalidate the conclusions of the analysis. For instance, robustness of 1 means that the conclusions are extendable to other situations provided that the data vary no more than 1 standard deviation. A variation of this magnitude can be expected just by chance, so one can have little confidence in asserting the conclusions for other situations.

However, greater robustness might be reached with larger sample sizes. This is because the $\chi^2$ test is sensitive to the sample size, as evidenced by Eqs (3) and (4). As an hypothetical experiment, let us proportionally double the observed $N_{ij}$ values in Table 3, recalculating both the $\chi^2$ test and the robustness analysis. The results are shown in Table 4.

As can be seen from the table, with doubled sample sizes, the conclusions are substantially robust in some but not all cases. For example, the association between gender and depression increases from robustness of less than 2, to 2.45 and 3.00 for pre-pandemic and pandemic-time periods, correspondingly. These values correspond to variation of the data up to 2.45 and 3.00 standard deviations, respectively. This implies reasonable confidence in applying these conclusions to other situations in this hypothetical case. The other two associations do not show large robustness for the hypothetically enlarged data set, suggesting that much larger samples would be needed. The conclusion from this hypothetical experiment is, in broad terms, that much larger samples are needed in order to obtain results that can be confidently

generalized to other times or populations. Only much larger samples would have substantial robustness to non-probabilistic uncertainty in the data.

### 5.3. Summary

The study of quantitative mental health assessments is traditionally conceptualized with two orthogonal axes. The first is focused on the relationship between questions about symptoms and diagnoses: We call it the *individual validity axis*. The second is the *collective validity axis*, solidly grounded in statistics and probability. Its primary concern is to allow generalization of results obtained by a sample, to the entire population. Much quantitative mental health research can be positioned in a virtual map by defining the utilized validity methods on one axis, and the inferential tools on the other.

We explored the concept of non-probabilistic Knightian uncertainty arising in other possible samples. We employed info-gap analysis of robustness to assess the impact of this Knightian uncertainty. Info-gap robustness is ontologically different from statistical uncertainty because it entails no probabilistic or statistical assumptions. Robustness to this type of deep Knightian uncertainty enables one to assess the degree of confidence in inference from a known situation to a hypothetical new situation that may differ fundamentally. Robustness to deep uncertainty is orthogonal both to the *individual validity axis* and the *collective validity axis*, in the sense that it does not necessarily affect either of these axes. For example, info-gap analysis is indifferent to the implemented validation method. In addition, when examining statistical inference, robustness analysis is indifferent to whether the null hypothesis is accepted or rejected.

The introduction of info-gap theory as a tool to assess deep uncertainty creates an additional axis that expands the domain of analysis in quantitative mental health assessments. The new opportunities opened by this approach were shown by using a $\chi^2$ test followed by an info-gap analysis of the results. The limitations of the approach were highlighted also, with suggestions about ways to overcome them.

The ability to measure robustness to deep uncertainty in the mental health realm is vital because the research area is based mainly on questionnaires that reflect the conditions of a specific population and time. The ability to measure robustness is important particularly in troubled and changing times. A recent example is the outbreak of Covid-19, an unexpected,

**Table 4. Associations between mental health disorders and different socio-demographic characteristics of simulated enlarged samples.** The real values appear without parentheses, while the values obtained using enlarged samples are in parentheses.

| | Socio-demographic characteristic | | Anxiety disorder | | Depression disorder | | Somatic disorder | |
|---|---|---|---|---|---|---|---|---|
| | | | Q0 | Q1* | Q0 | Q1* | Q0 | Q1* |
| 1 | Gender | $\chi^2$ statistic | | | **18.52 (37.05)** | **22.77 (45.55)** | | |
| | | Critical robustness | | | 1.40 (2.45) | 1.75 (3.00) | | |
| 4 | Marital status | $\chi^2$ statistic | 3.23 **(6.46)** | **5.92 (11.85)** | | | | |
| | | Critical robustness | 0.05 (0.30) | 0.20 (0.75) | | | | |
| 5 | Living with a child | $\chi^2$ statistic | | | | | **9.82 (19.64)** | 0.74 (1.47) |
| | | Critical robustness | | | | | 0.55 (1.25) | 0.55 (0.40) |

traumatic, and worldwide event that may have changed human behavior in several ways, including its impact on mental health. The analysis of robustness to deep uncertainty, implemented using info-gap theory, seems to be particularly suitable for this type of situation. In the present paper, we provide the basic methodological building blocks of the suggested approach through a simple example.

## Author Contributions

**Conceptualization:** Nataliya Rybnikova, Dani Broitman, Yakov Ben-Haim.

**Data curation:** Murielle Mary-Krause, Maria Melchior.

**Formal analysis:** Nataliya Rybnikova, Dani Broitman, Yakov Ben-Haim.

**Funding acquisition:** Dani Broitman, Maria Melchior, Yakov Ben-Haim.

**Investigation:** Nataliya Rybnikova, Dani Broitman, Yakov Ben-Haim.

**Methodology:** Nataliya Rybnikova, Dani Broitman, Yakov Ben-Haim.

**Project administration:** Dani Broitman.

**Software:** Nataliya Rybnikova.

**Supervision:** Dani Broitman, Maria Melchior.

**Validation:** Nataliya Rybnikova, Dani Broitman, Murielle Mary-Krause, Maria Melchior, Yakov Ben-Haim.

**Visualization:** Nataliya Rybnikova, Dani Broitman, Yakov Ben-Haim.

**Writing – original draft:** Nataliya Rybnikova, Dani Broitman, Murielle Mary-Krause, Maria Melchior, Yakov Ben-Haim.

**Writing – review & editing:** Nataliya Rybnikova, Dani Broitman, Murielle Mary-Krause, Maria Melchior, Yakov Ben-Haim.

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
