## [Decision Letter · Decision Letter 0]

20 Jul 2023

PONE-D-23-07703Uncertainty in the association between Socio-Demographic characteristics and Mental HealthPLOS ONE

Dear Dr. Broitman,

Thank you for submitting your manuscript to PLOS ONE. After careful consideration, we feel that it has merit but does not fully meet PLOS ONE’s publication criteria as it currently stands. Therefore, we invite you to submit a revised version of the manuscript that addresses the points raised during the review process.

I encourage the authors to carefully address the comments from the reviewers and in particular, those made by reviewer 2. I agree that the mothodology used should be discussed more in depth and especially the info gap model and its probabilistic assumptions.

We look forward to receiving your revised manuscript.

Kind regards,

Maurizio Fiaschetti

Academic Editor

PLOS ONE

Journal Requirements:

Could you therefore please include the title page into the beginning of your manuscript file itself, listing all authors and affiliations

"This paper is a research output of the project “COVID-19 and Mental Health – dealing with short and long-term Uncertainty” (COMHU). COMHU is a bi-national research project funded by the Ministry of Science and Technology (MOST) of the State of Israel, the Ministry of Europe and Foreign Affairs (MEAE) and the Ministry of Higher Education, Research and Innovation (MESRI) of France. The authors thank the funding institutions for their support." 

Reviewers' comments:

Reviewer's Responses to Questions

**Comments to the Author**

1. Is the manuscript technically sound, and do the data support the conclusions?

Reviewer #1: Yes

Reviewer #2: Partly

2. Has the statistical analysis been performed appropriately and rigorously? 

Reviewer #1: Yes

Reviewer #2: No

3. Have the authors made all data underlying the findings in their manuscript fully available?

Reviewer #1: No

Reviewer #2: No

4. Is the manuscript presented in an intelligible fashion and written in standard English?

Reviewer #1: No

Reviewer #2: No

5. Review Comments to the Author

Reviewer #1: This is a very interesting piece of work that highlights the importance of a "robustness

to deep uncertainty axis" as a third dimension in the consideration of psychometric information where, currently validity in terms of individual and generalisability robustness are generally established. Thus, it is a third dimension to take into account. The authors use a French epidemiological survey about mental health impacts of the Covid pandemic to explore their ideas. They develop a model of info-gap robustness to

mental health assessment, showing how the robustness to deep uncertainty axis interacts

with the other two axes. They highlight the contributions and the limitations of this

approach.

I have two suggestions to what seems a valid piece of work. First, to expand the description of "Knightian" uncertainty in the introduction. This concept is crucian to the argument. I had to explore extensively beyond the manuscript to begin to understand this and, had I not been reviewing the paper, might have given up.

Similarly, the authors might consider including the crux of their findings from Tempo in the abstract. This would help the reader to be carried through what is quite complex (for a non-psychometrician) writing. At the moment, the point of the paper is demonstrated only quite late in the manuscript.

Readers are unable to replicate the findings. Can the authors say more about data availability and make their code available?

Reviewer #2: The paper deals with the development of a new approach to mease robustness in order to deepen uncertainty in the test of association between socio-demographic factors and Mental Health. The analysis of deep uncerteinty is carried out using the concept of robustness in info-gap decision theory. Data collected in surveys carried out before and during the COVID-19 pandemic are analysed.

The paper is interesting and cover newsworthy topics from both, the methodological (robustness in info-gap decision theory) and the practical (association between socio-demographic characteritics) points of view. However it cannot be published in the current version. If the following changes are made, it may be suitable for publication. Please, in the revised version, highlight the changes with respect to the original submission.

GENERAL COMMENT

In general, some descriptions of the adopted methods are not clear and sufficient. Furthermore several choices about methodological aspects are not well justify and some of them are quite questionable. Furthermore, there are errors and inaccuracies in the notations used in the formulas.

SPECIFIC COMMENTS

The literature review about contributions on mental health issues in COVID-19 pandemic is not exhaustive and must be extended. Some examples of interesting papers on the topic are the following:

- Moreno, C., Wykes, T., Galderisi, S., Nordentoft, M., Crossley, N., Jones, N., Cannon, M., Correll, C.U., Byrne, L., Carr, S., et al. (2020). How mental  health  care  should  change  as  a  consequence  of  the  COVID‐19  pandemic. Lancet Psychiatry, 7, 813–824. https://doi.org/10.1016/S2215‐0366(20)30307‐2. 

- Msherghi, A., Alsuyihili, A., Alsoufi, A., Ashini, A., Alkshik, Z., Alshareea, E., Idheiraj, H., Nagib, T., Abusriwel, M., Mustafa, N., et al. (2021) Mental Health Consequences of Lockdown During the COVID‐19 Pandemic: A Cross-Sectional Study. Front. Psychol. 12, 605279. https://doi.org/10.3389/fpsyg.2021.605279. 

- Hervalejo, D., Carcedo, R.J., and Fernández-Rouco, N. (2020) Family and mental health during the confinement due to the COVID‐19 pandemic in Spain: The perspective of the counselors participating in psychological helpline services. J. Comp. Fam. Stud. 51, 399–416. https://doi.org/10.3138/jcfs.51.3‐4.014. 

- Bonnini S, Borghesi M (2022), Relationship between Mental Health and Socio‐Economic, Demographic and Environmental Factors in the COVID‐19 Lockdown Period‐A Multivariate Regression Analysis, Mathematics, 10(18), 3237. https://doi.org/10.3390/math10183237

Ahmed, M.Z., Ahmed, O., Aibao, Z., Hanbin, S., Siyu, L., Ahmad, A. (2020) Epidemic of COVID‐19 in China and associated psycho‐logical problems. Asian J. Psychiatry, 51, 102092. https://doi.org/10.1016/j.ajp.2020.102092.

page 4, section 2, 5 lines after the title: "We should note that we intentionally chose a simplified statistical technique in order to focus primarily on the robustness analysis...". I understand that the robustness analysis is the main contribution of the study but, in general, you should choose a suitable statistical technique according to the design of the study, the goals, the type of variables, the assumptions and the reliability, and not according to simplicity. Either the statistical technique is suitable or it is not. Please comment on this point and explain better.

page 5, 4th line from the top: you used a very unusual notation for p_ij, because usually, in matrix notation, the first subscriptm i represents the row and the second j the column...

page 5, from 4th line to formula (1): the explanation of the test and of the hypotheses is confused, just to say that you compare the marginal probability distributions of the two populations. Anyway, such definition of the problem is misleading because the test on the association between two categorical variables is conceptually different from the test on the comparison between the probability distributions of two populations. You should change and simplify the problem definition.

formula (1): p_12=p_22 is unuseful

in order to test the effects of socio-demografic fectors you should consider a suitable procedure that jointly take into account all the factors simultaneously. If you consider one factor at a time, you don't test the associations between variables net of the others. Comment on this choice of considering separate chi-square tests, one per socio-demographic factor.

page 5, formulas (3) and (4) and later in the paper: the same notation is used to represent the observed and the expected number of observations. Please, distinguish the two quantities.

page 5, 5th line from the bottom: replace "Let Qc(p) denote the value of the chi-square statistic at a level of significance p with one degree of freedom" with "Qc(p) represents the quantile of the chi-square distribution corresponding to a cumulative probability of p".

page 6, 1st row after formula (5): replace "accepted" with "not rejected"

page 6, from 5th line to the 2nd line the bottom, page 7, formula (6): you state that the info-gap uncertainty does not depend on any probabilistic assumptions but you base the info-gap model on the assumption that the observations are generated by a Poisson random variable. This is an evident contraddiction! What happens if the data are not Poisson distributed? Is there a general formulation that doesn't need the assumption of Poisson distribution?

page 7, 1st and 2nd line after formula (6): "The N_ij only take discrete values, but we are treating them as continuous variables for simplicity". Again simplicity drives your methodological choices. This is questionable because discrete data should imply the use of discrete variables. You should better justify this choice and motivate the fact that it doesn't adversely affect the reliability of results.  

page 8, 3rd line from the top: why do you consider the inverse function? you should justify this choice and motivate the fact that it doesn't affect the reliability of results.

page 8, 2nd line to the title of section 3: please clearly indicate which tables are you specifically talking about

page 10, 10th line to the bottom: why do you choose the 85th percentile? please justify this choice

6. PLOS authors have the option to publish the peer review history of their article (what does this mean?). If published, this will include your full peer review and any attached files.

Reviewer #1: **Yes: **PB Jones

Reviewer #2: No

---

## [Author Response · Author response to Decision Letter 0]

11 Sep 2023

We addressed all the issues raised by the reviewers, detailed in a separate file called "Response To Reviewers"

---

## [Decision Letter · Decision Letter 1]

11 Oct 2023

PONE-D-23-07703R1Uncertainty in the association between Socio-Demographic characteristics and Mental HealthPLOS ONE

Dear Dr. Broitman,

Thank you for submitting your manuscript to PLOS ONE. After careful consideration, we feel that it has merit but does not fully meet PLOS ONE’s publication criteria as it currently stands. Therefore, we invite you to submit a revised version of the manuscript that addresses the points raised during the review process.

We look forward to receiving your revised manuscript.

Kind regards,

Maurizio Fiaschetti

Academic Editor

PLOS ONE

Journal Requirements:

**Additional Editor Comments:**

The manuscript has addressed all but one point that Reviewer 2 has kindly highlighted. I encourage the authors to do so and resubmit.

Reviewers' comments:

Reviewer's Responses to Questions

**Comments to the Author**

1. If the authors have adequately addressed your comments raised in a previous round of review and you feel that this manuscript is now acceptable for publication, you may indicate that here to bypass the “Comments to the Author” section, enter your conflict of interest statement in the “Confidential to Editor” section, and submit your "Accept" recommendation.

Reviewer #1: All comments have been addressed

Reviewer #2: (No Response)

2. Is the manuscript technically sound, and do the data support the conclusions?

Reviewer #1: Yes

Reviewer #2: Yes

3. Has the statistical analysis been performed appropriately and rigorously? 

Reviewer #1: Yes

Reviewer #2: Yes

4. Have the authors made all data underlying the findings in their manuscript fully available?

Reviewer #1: Yes

Reviewer #2: Yes

5. Is the manuscript presented in an intelligible fashion and written in standard English?

Reviewer #1: Yes

Reviewer #2: Yes

6. Review Comments to the Author

Reviewer #1: The authors have done a very good job at addressing both sets of reviewer comments. The description of the phenomenon in question is now very useful and the the ms is much more accessible. I like the catastrophe-like figure. I hope the work gets some traction in the field.

Reviewer #2: All the suggestions of the previous review have been taken into account except the following:

"page 5, 5th line from the bottom: replace "Let Qc(p) denote the value of the chi-square statistic at a level of significance p with one degree of freedom" with "Qc(p) represents the quantile of the chi-square distribution

corresponding to a cumulative probability of p".

Of course, you are not obliged to use the suggested sentence, but you must change the original one because it is not correct. The value of a statistic is only function of data, neither of the significance level nor of the degrees of freedom. The degrees of freedom and the value of p, concerns a probability distribution. Your sentence makes no sense.

In my opinion, if also this point is addressed, the paper is suitable for publication in Plos One.

7. PLOS authors have the option to publish the peer review history of their article (what does this mean?). If published, this will include your full peer review and any attached files.

Reviewer #1: **Yes: **Peter B. Jones

Reviewer #2: No

---

## [Author Response · Author response to Decision Letter 1]

17 Oct 2023

A detailed table with responses to the reviewer's comments is submitted as a separated file.

---

## [Decision Letter · Decision Letter 2]

24 Oct 2023

PONE-D-23-07703R2Uncertainty in the association between Socio-Demographic characteristics and Mental HealthPLOS ONE

Dear Dr. Broitman,

Thank you for submitting your manuscript to PLOS ONE. After careful consideration, we feel that it has merit but does not fully meet PLOS ONE’s publication criteria as it currently stands. Therefore, we invite you to submit a revised version of the manuscript that addresses the points raised during the review process.

 The authors are strongly advised to check the formatting of the manuscript in order to allow the reviewers to do their job. In particular, they should double check all the equations and make sure that nothing is lost in the conversion of the manuscript to different file formats.

We look forward to receiving your revised manuscript.

Kind regards,

Maurizio Fiaschetti

Academic Editor

PLOS ONE

Journal Requirements:

Reviewers' comments:

Reviewer's Responses to Questions

**Comments to the Author**

1. If the authors have adequately addressed your comments raised in a previous round of review and you feel that this manuscript is now acceptable for publication, you may indicate that here to bypass the “Comments to the Author” section, enter your conflict of interest statement in the “Confidential to Editor” section, and submit your "Accept" recommendation.

Reviewer #2: All comments have been addressed

2. Is the manuscript technically sound, and do the data support the conclusions?

Reviewer #2: Yes

3. Has the statistical analysis been performed appropriately and rigorously? 

Reviewer #2: Yes

4. Have the authors made all data underlying the findings in their manuscript fully available?

Reviewer #2: Yes

5. Is the manuscript presented in an intelligible fashion and written in standard English?

Reviewer #2: Yes

6. Review Comments to the Author

Reviewer #2: In my opinion the paper is now almost suitable for publication. To makle it suitable, you should solve a formal problem. In equations 7 and 8, and in parts of the text, I see rectangles but probably this is not the right symbol the authors wanted to choose...please check and correct.

7. PLOS authors have the option to publish the peer review history of their article (what does this mean?). If published, this will include your full peer review and any attached files.

Reviewer #2: No

---

## [Author Response · Author response to Decision Letter 2]

27 Oct 2023

We addressed the concerns raised by the reviewer in our revised version, rewriting some equations and mathematical expressions in the manuscript. We hope the editor will find this version suitable for publication in the journal.

---

## [Decision Letter · Decision Letter 3]

7 Nov 2023

Uncertainty in the association between Socio-Demographic characteristics and Mental Health

PONE-D-23-07703R3

Dear Dr. Broitman,

We’re pleased to inform you that your manuscript has been judged scientifically suitable for publication and will be formally accepted for publication once it meets all outstanding technical requirements.

Kind regards,

Maurizio Fiaschetti

Academic Editor

PLOS ONE

Additional Editor Comments (optional):

Reviewers' comments:

Reviewer's Responses to Questions

**Comments to the Author**

1. If the authors have adequately addressed your comments raised in a previous round of review and you feel that this manuscript is now acceptable for publication, you may indicate that here to bypass the “Comments to the Author” section, enter your conflict of interest statement in the “Confidential to Editor” section, and submit your "Accept" recommendation.

Reviewer #2: All comments have been addressed

2. Is the manuscript technically sound, and do the data support the conclusions?

Reviewer #2: Yes

3. Has the statistical analysis been performed appropriately and rigorously? 

Reviewer #2: Yes

4. Have the authors made all data underlying the findings in their manuscript fully available?

Reviewer #2: Yes

5. Is the manuscript presented in an intelligible fashion and written in standard English?

Reviewer #2: Yes

6. Review Comments to the Author

Reviewer #2: All my comments have been taken into account by the authors in the revised manuscript, including the last one concerning the formal problem of the wrong symbol. The paper is now suitable for publication.

7. PLOS authors have the option to publish the peer review history of their article (what does this mean?). If published, this will include your full peer review and any attached files.

Reviewer #2: No

---

## [Editor Report · Acceptance letter]

10 Nov 2023

PONE-D-23-07703R3 

Uncertainty in the association between Socio-Demographic characteristics and Mental Health 

Dear Dr. Broitman:

I'm pleased to inform you that your manuscript has been deemed suitable for publication in PLOS ONE. Congratulations! Your manuscript is now with our production department. 

Kind regards, 

on behalf of

Dr. Maurizio Fiaschetti 

Academic Editor

PLOS ONE